# The Paracrine Role of Endothelial Cells in Bone Formation via CXCR4/SDF-1 Pathway

**DOI:** 10.3390/cells9061325

**Published:** 2020-05-26

**Authors:** Tal Tamari, Rawan Kawar-Jaraisy, Ofri Doppelt, Ben Giladi, Nadin Sabbah, Hadar Zigdon-Giladi

**Affiliations:** 1Laboratory for Bone Repair, Rambam Health Care Campus, Haifa 3109601, Israel; frumzi@gmail.com (T.T.); ofridoppelt@gmail.com (O.D.); 2The Ruth and Bruce Rappaport Faculty of Medicine, Technion—Israel Institute of Technology, Haifa 3200003, Israel; bengiladi88@gmail.com (B.G.); nadin.tech@gmail.com (N.S.); 3The Maurice and Gabriela Goldschleger School of Dental Medicine, Tel Aviv 69978, Israel; rawankawar@gmail.com

**Keywords:** endothelial cells, mesenchymal stem cells, bone, angiogenesis, paracrine, SDF-1, CXCR4

## Abstract

Vascularization is a prerequisite for bone formation. Endothelial progenitor cells (EPCs) stimulate bone formation by creating a vascular network. Moreover, EPCs secrete various bioactive molecules that may regulate bone formation. The aim of this research was to shed light on the pathways of EPCs in bone formation. In a subcutaneous nude mouse ectopic bone model, the transplantation of human EPCs onto β-TCP scaffold increased angiogenesis (*p* < 0.001) and mineralization (*p* < 0.01), compared to human neonatal dermal fibroblasts (HNDF group) and a-cellular scaffold transplantation (β-TCP group). Human EPCs were lining blood vessels lumen; however, the majority of the vessels originated from endogenous mouse endothelial cells at a higher level in the EPC group (*p* < 01). Ectopic mineralization was mostly found in the EPCs group, and can be attributed to the recruitment of endogenous mesenchymal cells ten days after transplantation (*p* < 0.0001). Stromal derived factor-1 gene was expressed at high levels in EPCs and controlled the migration of mesenchymal and endothelial cells towards EPC conditioned medium in vitro. Blocking SDF-1 receptors on both cells abolished cell migration. In conclusion, EPCs contribute to osteogenesis mainly by the secretion of SDF-1, that stimulates homing of endothelial and mesenchymal cells. This data may be used to accelerate bone formation in the future.

## 1. Introduction

During embryogenesis and the post-natal period, vascularization precedes osteogenesis and is a prerequisite for bone formation. An intimate physical proximity exists between blood vessels, osteoblasts, and osteoclasts. Endothelial cells function is not limited to creating a barrier between the blood and the vascular smooth muscle. They also control vasodilatation and vasoconstriction, regulate vessel permeability, and secrete an array of soluble mediators, such as: fibroblast growth factor (FGF), interleukin-1 (IL-l), interleukin-6 (IL-6), and colony stimulating factors (CSFs), that communicate mostly with immune cells, but may also regulate osteoblasts’ and osteoclasts’ function [1,2]. The proximity of endothelial cells to osteoblasts and osteoclasts, and the production of bone regulating substances suggest endothelial cells, may play a key role in bone cell development and activity.

Bone has the ability to undergo regeneration after injury, however, successful regeneration relies on coordination between osteoprogenitors and blood vessels forming cells [3]. Therefore, reduction in the amount or function of osteoprogenitors and endothelial cells may lead to fibrous healing instead of bone fill [4]. For example, large bone defects or nonunion fractures, that often involve inadequate angiogenesis, and bone forming cells fail to heal and require bone grafting procedures to achieve bone fill. Autogenous bone block is the gold standard technique to treat large bone defects [5]. However, this technique is associated with many limitations, such as insufficient available bone at the donor site and lack of vascularity. Thus bone blocks can lead to significant volumetric reduction, and most of the cells in the block do not survive [6]. Other strategies have been developed as alternatives to autologous bone block, including those relying on the osteoconductive or osteoinductive capabilities of an implanted tissue (such as allograft or demineralized bone matrix) or synthetic material (such as bio ceramics in different forms and compositions) [7,8]. Growth factors that participate in mesenchymal stem cells (MSCs) recruitment and MSC osteogenic differentiation have been incorporated into scaffolds, demonstrating promising results [9]. Cell-based therapies were investigated, mainly using bone marrow or adipose derived MSCs [10]. One of the limiting factors for all of the above mention strategies is inadequate vascularization to the graft [11,12].

Treatment with endothelial progenitor cells (EPCs) was investigated as a tool for bone therapy, since angiogenesis plays a pivotal role in successful bone regeneration and EPCs participate in pre- and post-natal angiogenesis [13]. EPCs are a minor population of mononuclear cells found in the peripheral blood, that are the precursors of endothelial cells. EPCs are involved in both revascularization (new blood vessel formation from pre-existing ones) and neovascularization (new blood vessel formational without preexisting vessels) [11,14]. There are two subtypes of EPCs, early EPCs with myeloid\hematopoietic characteristics [15,16] and late EPCs [17]. Early EPCs contribute to angiogenesis via a paracrine effect while late EPCs are directly incorporated into neovessels [18,19].

EPCs can promote the neovascularization of tissue-engineered bones, and further improve osteogenesis and bone reconstruction. Preclinical studies in rat and sheep models with rat, sheep, and human EPCs have demonstrated significant enhancement in intra-membranous and endochondral bone regeneration, following the local transplantation of blood derived EPCs into large bone defects. In sheep tibia [20] (endochondral bone), 3.5mm critical size defects were regenerated with new bone, following local transplantation of autologous EPCs, achieving complete bridging of the defects (7 out of 8 animals, compared to none in the control). In the rat calvaria (intra-membranous bone) model, 5mm new bone height was achieved compared with 2.5mm new bone formation in the control [21]. Moreover, in a 5-month follow-up study, no pathological conditions were observed in the DNA of the cells or target organs of transplanted nude rats [22].

Despite a significant body of evidence showing that EPCs contribute to bone regeneration, the mechanism of action remains unclear. The most common belief is that EPCs create a vascular network, through which nutrient and cells can be delivered to the tissue during healing and regeneration. We hypothesize that EPCs participate in angiogenesis through direct mechanism, where they actively participate in forming blood vessels’ walls, and in paracrine mechanism by recruiting host angiogenic cells to the regenerated site. They also participate in osteogenesis in paracrine way, by recruiting host osteogenic cells. The aim of this research was to shed light on the direct and indirect pathways of the EPCs in bone formation. We chose a subcutaneous ectopic bone formation model, since no osteoprogenitors are present at this site. Thus, ectopic bone formation is dependent on the ability of EPCs to differentiate into osteoprogenitors, or to recruit endogenous osteoprogenitor cells.

## 2. Materials and Methods

The study protocol was approved by the Committee for the Supervision of Animal Experiments at the Technion’s Ruth and Bruce Rappaport Faculty of Medicine (approval number IL0810712), and by the Helsinki committee for human experiments of the Rambam Health Care Campus (Helsinki number 0397-12–RMB).

### 2.1. Isolation, Expansion and Characterization of EPC

Blood was collected from post trauma hospitalized orthopedic patients, and healthy donors aged 19–44 years. Donors were asked to sign a consent form before participating in the study (Helsinki number 0397-12–RMB). Blood was transferred to 50mL heparinized tubes (Biological Industries Ltd., Kibbutz Beit Haemek, Israel) and diluted to a 1:1 ratio with phosphate buffered saline (PBS) (Biological Industries Ltd.) The diluted blood was added to another 50mL tube that contained lymphoprep™ (Fresenius Kabi Norge AS, Axis-Shield PoC AS, Oslo, Norway) in a ratio of 1:1:1, and centrifuged (Hermle Z 300, ©HERMLE Labortechnik GmbH, Wehingen, Germany) at 750 xg/slow, for 30 min at room temperature. The buffy coat was removed with a pipette to a 50mL tube and washed twice with PBS. Pelleted cells were suspended in an endothelial growth medium (EGM-2). The medium was composed of endothelial growth medium (EGM-2) (EGM-2MV SingleQuote; Clonetics: Lonza, Basel, Switzerland), containing 20% fetal bovine serum (FBS), penicillin-streptomycin (Biological Industries Ltd.), and supplemented with endothelial growth medium (EGM-2MV SingleQuote; Clonetics, Lonza, Basel, Switzerland). Cells were seeded on six-well plates coated with fibronectin (Biological Industries Ltd.) and cultured at 37 °C with humidified 95% air/5% CO_2_. After 4 days in culture, non-adherent cells were washed with PBS, and fresh EGM-2 medium was applied. Cells were cultured until the first colonies of late EPCs appeared, usually after 2–3 weeks. Late EPCs were fed three times weekly and split at confluence (~80%) using trypsin (Biological Industries Ltd.) [21]. Human EPCs were characterized using flow cytometry: 2 × 10^5^ cells were suspended in 50 μL Fluorescence-activated cell sorting (FACS) buffer (PBS containing 0.5% FBS) and stained with 0.2 mg/mL of anti-human antibodies: CD31 (clone 1D2-1A5 LifeSpanBioSciences, Seattle, WA, USA), CD34 (clone 581, BD Biosciences, San Jose, CA, USA), CD45 (clone HI30, BD Biosciences, San Jose, CA, USA), VEGFR-2 (clone #89106, R&D SYSTEMS, Minneapolis, MN, USA) and CD14 (clone M5E2, BD Biosciences, San Jose, CA, USA). OneComp eBeads (ThermoFischer Scientific, Waltham, MA, USA) were stained with 1 μL of each different fluorochrome then used as single-color compensation controls. Cells were analyzed using cyan flow cytometry (Beckman Coulter, Brea, CA, USA). Single cell data were analyzed using FlowJo, LLC software (BD Biosciences, San Jose, CA, USA) that gave a percentage for positive cells. Early EPCs were positive for CD31, CD45, VEGFR-2 and CD14. Late EPCs were positive for: CD31, CD34 and negative for CD14 and CD45 [23].

### 2.2. HNDF Culture

Human neonatal dermal fibroblasts (HNDF) (Lonza, Basel, Switzerland) are primary cells derived from the dermis of normal human neonatal foreskin. Cells were cultured in DMEM supplemented with 10% FBS, 1% l-Glutamine and 1% penicillin-streptomycin (Biological Industries Ltd.) Medium was changed biweekly and cells were split at confluence (~80%) using Trypsin (Biological Industries Ltd.).

### 2.3. MSCs Culture

Human bone marrow mesenchymal stem cells (MSCs) were cultured as previously described [24]. Cells were cultured in DMEM and supplemented with 10% FBS, 1% l-Glutamine and 1% penicillin-streptomycin (Biological Industries Ltd.) Medium was changed three times a week and cells were split at confluence (~80%) using Trypsin (Biological Industries Ltd.)

### 2.4. DII Staining

Cells were labelled with fluorescent dye before implantation. Cells were suspended in 1ml PBS, containing 25 µl DII stain (ThermoFischer Scientific, Waltham, MA, USA), then incubated for 30 min. Cells were centrifuged and washed twice before implantation.

### 2.5. In Vivo Ectopic Bone Model

Subcutaneous transplantation allows one to track the implanted cells and newly formed blood vessels in vivo, which cannot be done in an orthotopic bone model.

### 2.6. Scaffold Preparation

Synthetic β-Tricalcium phosphate (β-TCP) (0.6–1 mm grain size, ~80% porosity and 100–200 µm pore size, Poresorb-TCP^®^ (Lasak Ltd., Prague, Czech Republic) were used based on our previous results [25], and due to its resorbable characteristic during bone regeneration; 0.1gr β-TCP was coated with 60 µl fibronectin and 150 µl PBS. After 1 h incubation, 2 × 10^5^ EPCs or 2 × 10^5^ HNDF were added to each β-TCP coated with fibronectin mass; then, 150 µl fibrinogen (Sigma-Aldrich, Saint Louis, MO, USA) and 75 µl thrombin (Sigma-Aldrich) were added and mixed together.

### 2.7. Subcutaneous Cell Transplantation

Forty-two female nude mice (10 weeks, 25 g) were anaesthetized with 2% Isoflurane USP Terrell™ (Piramal Critical Care Inc., Bethlehem, PA, USA), in 100% oxygen. Three to five small subcutaneous pouches were made on the back of each mouse (5 mm each). Each mouse was treated with one of the following groups: (i) EPC (*n* = 18); (ii) HNDF (*n* = 6); and, (iii) β-TCP (control, *n* = 18). Following insertion of the scaffold into the subcutaneous pouches, flaps were repositioned and sutured. Mice were kept in separate cages and fed rat chow. Mice were sacrificed at ten days, three weeks, and eight weeks, by CO_2_ asphyxiation.

### 2.8. Dextran Preparation and Injection

Fluorescein isothiocyanate-dextran (Sigma-Aldrich) was dissolved in PBS, to a concentration of 10 mg/mL; 0.2 mL of the dissolved dextran was injected into the tail vein in order to label functional blood vessels in green. After sacrifice, biopsies were taken and fixed immediately in 4% paraformaldehyde (Bio-Lab Ltd., Jerusalem, Israel) for 10–20 min, then washed with PBS. Functional blood vessels were visualized with LSM 510 Zeiss laser confocal system (Zeiss, Oberkochen, Germany). A 2 × 2 mm sample from each transplant was excised and embedded in 1% agarose gel, and then a 3D visualization of functional vessels was performed using a Lightsheet Z.1 microscope (Zeiss). Blood vessel density was quantified and calculated by dividing blood vessel volume by tissue volume, using IMARIS software v8.3 (Zurich, Switzerland).

### 2.9. Histological Preparation

Specimens were fixed with 4% paraformaldehyde (Bio-Lab Ltd., Jerusalem, Israel) underwent decalcification in 10% EDTA (Sigma-Aldrich) for 3 days, were embedded in paraffin, sectioned (5 μm), and were stained with H&E.

### 2.10. Immunohistochemistry

Each section was blocked with Background Block Buster (Innovex Bioscience Inc., Richmond, CA, USA) for 30 min, rinsed twice with PBS for 5 min, and stained with anti-mouse CD73 antibody (NBP1-85740, Novusbio, Centennial, CO, USA), anti-mouse CD31 antibody (Mouse/Rat CD31/PECAM-1, R&D systems, Minneapolis, MN, USA), HNA (Human Nuclear Antigen, clone NM95, Scytek, Logan, UT, USA), and anti-human CD31 (clone JC70, ZYTOMED, Berlin, Germany), for 60 min at room temperature. After rinsing for three times, slides were stained with HRP (ZYTOMED, Berlin, Germany) for 30 min, rinsed, and stained with DAB (ThermoFischer Scientific, Waltham, MA, USA) for 5–8 min. After rinsing again, the slides were stained with Hematoxylin (10% Hematoxylin, 90% distilled water) for 30–60 s, and washed with distilled water. Ten random fields from each slide were captured by a microscope Olympus CX31camera (Olympus, Tokyo, Japan) and used to quantify immunostaining with Image-Pro premier software (Media Cybernetics, Rockville, MD, USA).

### 2.11. Double Staining Immunohistochemistry

Slides were subjected to double staining immunohistochemistry, to detect the proximity between human and mouse antigens within the mouse’s subcutaneous implants. Slides were stained with Human Nuclear Antigen (HNA); (Scytek) and then restained with anti-mouse CD73 antibody (Novusbio).

### 2.12. EPC Conditioned Medium (EPC-CM) Preparation

One million human EPCs were cultured in EGM-2 (Lonza) with 20% FBS, until 80% confluence. After incubation for 48h, 10 mL supernatant was collected and centrifuged to remove cells (250× *g*, 5 min), then concentrated using a centrifugal filter (Merck Millipore Ltd., Tullagreen, Ireland) for 10 min in 2500× *g*. EPC-CM was compared to EGM-2 with 20% FBS

### 2.13. RT-PCR

RNA was extracted from pellets of EPCs and buffy coat cells obtained from two donors; 1 µg RNA from each sample was taken to reverse transcription reaction. Next, cDNA was generated with High-Capacity cDNA Reverse Transcription Kit (Thermo Fisher Scientific, Waltham, MA, USA). A quantitative real-time polymerase chain reaction (qPCR) was performed using real time PCR (Biometra Analytik, Jena, Germany) and Fast SYBR™ Green Master Mix (Applied Biosystems™, Foster City, CA, USA). An angiogenic kit containing 96 angiogenic primers (angiogenesis H96, Bio Rad, Hercules, CA, USA) was used and the results were normalized to generate fold change for each gene using the ΔΔCt method [26].

### 2.14. Migration Assay

The migration assay was performed with a 8 μm pore size millipore chamber (Millicell^®^, Darmstadt, Germany); 2.5 × 10^3^ MSCs or EPCs were seeded on the top of a porous membrane in 200 μL DMEM or EGM-2 (respectively), with 0.5% FBS. The lower chamber was filled with EGM-2, EPC-CM, or EGM-2+ 5ng/mL SDF-1 (stromal cell-derived factor 1) (PeproTech Asia Ltd., Rehovot, Israel). AMD3100 (Sigma-Aldrich) was added to the cells to block SDF-1 receptor. After overnight incubation, the cells were stained with crystal violet solution (Sigma-Aldrich) and the number of cells that migrated to the lower side of the membrane was quantified. The assay was performed in triplicates.

### 2.15. Statistical Analysis

SPSS v25 (IBM Inc., NY, USA) statistical software was used. Descriptive statistics, including means, medians, ranges, and standard deviations (SD), were initially calculated. Differences between groups were determined by a Kruskal–Wallis test with adjustments for multiple comparisons. Comparisons between control and select groups were performed using non-parametric Mann–Whitney U test. Comparisons between two groups with normal distribution were performed using *t*-test. The significance level was set at 5%.

## 3. Results

### 3.1. Human EPCs Enhance Angiogenesis and Mineralization in An Ectopic Bone Model

The role of EPCs in osteogenesis and angiogenesis was tested in an ectopic bone model. The microenvironment in the subcutaneous niche lacks bone cytokine and bone forming cell, therefore allowing for relatively controlled in vivo experimental bone formation. Initially, we examined whether EPCs transplantation stimulated angiogenesis and osteogenesis in this model, and compared the results to scaffold without cell (βTCP group) and human fibroblasts (HNDF group). Ten days following transplantation, mice were sacrificed, and newly formed functional vessels were demonstrated by dextran in the subcutaneous transplants. According to light sheet and confocal microscopy, blood vessels were scarce in the βTCP and HNDF groups, whereas in the EPCs group, a network of vessels was found. Quantitative analysis demonstrated a tenfold increase in blood vessel density in the EPC compared to in the βTCP and HNDF groups (*p* < 0.0001, Figure 1A).

In order to detect mineralization foci in the subcutaneous transplants, samples were extracted eight weeks after transplantation and prepared for histological analysis. The area of mineralized tissue was 77.14 ± 25.63 µm^2^ in the EPC samples (n = 9). However, mineralization was minimal in βTCP (n = 6) (*p =* 0.003) and HNDF samples (n = 6) compared to EPC, *p =* 0.001 (Figure 1B). These results indicate that the subcutaneous transplantation of EPCs enhanced angiogenesis and mineralization. Therefore, the ectopic bone model can be used to investigate the role of EPCs in these processes.

### 3.2. EPCs into Newly Formed Vessels in the Ectopic Transplants

An identification of EPCs in the ectopic transplants was performed using fluorescent microscope. Human EPCs and HNDF were labelled with fluorescent dye before transplantation. Human EPCs and HNDF were detected in the transplants, up to eight weeks after cell transplantation (Figure 2A). In order to localize the transplanted cells in the tissue, histological slides were stained with human specific antibodies: HNA (that stains the nucleus) and anti-human CD31. Both of the staining methods revealed integration of human EPCs and HNDF in the tissue; furthermore, the incorporation of EPCs into blood vessel walls was noticed (Figure 2B,C).

### 3.3. EPCs Recruit Endogenous CD31 and CD73 Cells in Vivo

Since the majority of the blood vessels in the transplants were negative to human antigens, our next step was to evaluate the effect of EPC on the recruitment of endogenous endothelial cells. In the EPC group, anti-mouse CD31 staining increased from the tenth day to three weeks (*p* = 0.002), but remained stable in the HNDF and βTCP groups *p* = 0.08. Intergroup comparison revealed higher mouse CD31 staining in the EPC at ten days (*p* < 0.01) and three weeks (*p* < 0.001), compared to the other groups (Figure 3A). Taken together, these results indicate that transplanted EPCs recruit endogenous mouse endothelial cells, which in turn play an active role in blood vessel formation.

In order to evaluate the osteogenic paracrine effect of EPCs, histological slides were stained with MSC surface marker, anti-mouse CD73. The stained area was measured using Image Pro software (Rockville, MD, USA). In the EPC group, CD73 increased from ten days to 3 weeks (*p* < 0.00001); while in the other groups, CD73 levels were stable. At three weeks, CD73 levels were higher in the EPCs than the other groups (*p* < 0.0001, EPCs vs. HNDF and βTCP) (Figure 3B). A double staining immunohistochemistry method for HNA and anti- mouse CD73 showed spatial proximity between transplanted EPCs and recruited mouse CD73 cells (Figure 3C). These results indicated that transplanted EPCs stimulate recruitment of mouse CD73^+^ cells during the three weeks after cell transplantation, that might have contributed to the mineralization of the tissue in the ectopic transplant.

### 3.4. SDF-1 in EPC Conditioned Medium (EPC-CM) Stimulate Migration of EPCs and MSCs In Vitro

In the ectopic bone model, we found that the main role of EPCs in angiogenesis and mineralization was by paracrine effect. To strengthen these results, a transwell migration assay was performed analyzing the migration of EPCs and MSC towards EPC conditioned medium (EPC-CM) (Figure 4A). The migration of MSCs towards EPC-CM was three folds higher compared with EGM-2 (*p =* 0.023). Similarly, EPCs migration doubled towards the EPC-CM (*p =* 0.011) (Figure 4B).

To understand how EPCs recruit endogenous cells (CD31+ and CD73+ cells) in vivo and stimulate migration of the same cells in vitro, we analyzed the mRNA levels of angiogenic and osteogenic genes in EPCs. Gene expression analysis revealed high levels of angiogenic genes, such as: *FGF*, *Kinase insert domain receptor (KDR)/vascular endothelial growth factor(VEGFR)*, *CXCL12/SDF-1*; and osteogenic genes, such as: *Platelet-derived growth factor subunit B (PDGF)* and *Bone morphogenetic protein 4 (BMP-4)*, compared to genes expressed by the mononuclear cells in the buffy coat of the same donors (Figure 5A). Using pathway analysis software, we detected several genes involved in cell mobilization, including: *SDF-1*, *CCL2*, and *PDGF*.

The migration of EPC and MSC towards EGM-2 supplemented with recombinant SDF-1 was higher than EGM-2 (*p =* 0.001, *p =* 0.0002 respectively), and comparable with migration towards EPC-CM. Likewise, blocking of SDF-1 receptor on MSC and EPC (using AMD3100) significantly reduced cell migration towards EPC-CM (*p* < 0.0001) (Figure 5B). Overall, these results suggest that SDF-1 is one of the pivotal molecules in EPC-CM modulating ECP and MSC migration.

## 4. Discussion

Many researchers have shown endothelial progenitor cells to be effective in cell-based therapies to improve vasculogenesis/angiogenesis, for a variety of therapeutic applications [27,28]. Other studies demonstrated that local application of EPCs into bone defects resulted in a significant increase in bone formation [20,29,30]. The accumulated evidence supports the contribution of EPCs to bone formation; nevertheless, the mechanisms of action remains unclear. Previous studies showed that endothelial cells produce and secrete soluble factors that may influence osteoblasts and osteoclasts functions [2,31,32,33,34,35,36], but there is only a poor understanding of the interrelationships between angiogenesis, bone formation, and bone remodeling in the maintenance of skeletal homeostasis.

Human EPCs were isolated from peripheral blood of healthy donors and hospitalized patients from the orthopedic department. All donors were healthy, without chronic diseases or medications, thus, we assume that the function of their cells is similar. EPCs isolation from healthy donors is challenging, due to their minimal number in the circulating blood. Previous studies isolated and cultured EPCs from blood, and were able to achieve sufficient cells for in vivo transplantation only in 52 percent of patients [37]. In an attempt to increase the odds to isolate EPCs from the circulation, blood was drawn from orthopedic patients after bone trauma. Shortly after, musculoskeletal traumatic event EPCs were mobilized into the circulation, therefore their levels are higher [38]. Alterations in EPCs genotype and phenotype after bone trauma were not studied, however it is possible to find slight changes in the cells between unrelated donors [23].

In this study, we investigated the role of endothelial progenitors in angiogenesis and osteogenesis, in a subcutaneous nude mouse model. According to the results, EPCs increased angiogenesis and osteogenesis by incorporating themselves into newly formed blood vessels, and mainly by recruiting resident MSCs and EPCs to the bone forming site. Prior to investigating the role of EPCs in angiogenesis and osteogenesis, we had to confirm that transplantation of EPCs into subcutaneous pouches selectively enhanced angiogenesis and mineralization. Indeed, mineralization foci were found even three weeks after EPCs transplantation, and was rarely found in the β-TCP and HNDF groups. Blood vessel density was also higher in the EPCs compared with β-TCP and HNDF groups. These results suggest that a subcutaneous model is suitable to investigate the role of EPCs in angiogenesis and osteogenesis.

The ectopic bone formation model refers to the ossification of tissues away from their original ossification site, which is to be distinguished from orthotopic bone formation, referring to studies in which bone is formed in its correct anatomical location. Of utmost importance, subcutaneous model lacks the naturally bone-forming stem cells within the dermal environment [39], which is a great advantage in the present study, since it enables us to evaluate the direct or indirect ability of EPCs to stimulate angiogenic and osteogenesis. Subcutaneous, intramuscular and kidney implantation are the most common ectopic bone formation models [40]. The subcutaneous implantation is the easiest and can be held on almost every mammalian animal. Mice are preferable and the most widely used because of their low cost, loose skin folds that can embrace large implants, and availability of immune-deficient mice that help prevent implant rejection by the host [39,41].

For a transplantation of EPCs, cells were seeded onto β-TCP scaffold. β-TCP is a synthetic scaffold consisting of a porous form of calcium phosphate that resembles human cancellous bone in structure and compositions [42]. β-TCP contains approximately 39% calcium and 20% phosphorus by weight, similar to the natural mineral content in bone [43]. It is a highly porous void-filler that is composed of 90% interconnected void space with a broad range of pore sizes, and was developed to mimic the trabecular structure of cancellous bone [8]. β-TCP is considered to be a good osteoconductive scaffold, to which circulating cells (e.g., MSCs, EPCs) can migrate, adhere, and differentiate (e.g., into osteoblast) [44,45]. The micro porous surface of β-TCP facilitates the anchorage of proteins and cells to the graft [46]. Its osteoconductive properties are due to the adherence of osteoblasts and deposition of organic and inorganic bone components within the pores of the scaffold [47]. It is also nontoxic, immunologically inert, non-carcinogenic and non-teratogenic [48]. β-TCP is biodegradable and permits the ingrowth of cells and vessels [49]. β-TCP showed superior osteoconductivity and lower inflammatory response in comparison to collagen and bovine derived xenofragt [25]. Moreover, fibronectin coated β-TCP enables EPCs adherence to the scaffold and was successfully used to treat bone defects [50].

Following the validation of the ectopic nude mouse model, showing EPC enhanced mineralization and angiogenesis in the subcutaneous niche, the next step was to determine the mechanism. We started by studying the direct engraftment of the transplanted cells and tracked the DII labeled cells ex vivo. Fluorescent signal was detected up to eight weeks after EPC and HNDF transplantation, with noticed reduction over time. This decrease in the fluorescent signal may be due to the nature of the lipophilic cell tracking dye that decreases in every cell division [51]. An additional explanation for the faint dye could be clearance by the host immune system [52] or apoptosis [53]. Hur et al. [17] stained early EPCs with a similar fluorescence DII and found that most of the cells died within two weeks. To validate the engraftment of human EPC and HNDF and localize the cells, tissue was stained with human specific antibodies for HNA and CD31. While EPC were integrating into blood vessel walls, HNDF were detected in the connective tissue surrounding β-TCP particles. Similarly Suh et al. demonstrated incorporation of human EPCs into new blood vessels via specific CD31 anti human [54]. Moreover, in a myocardial ischemia rat model, EPCs were injected intravenously. Seven days later, transplanted EPCs accumulated in the ischemic area and were incorporated into newly formed blood vessels [55]. However, most of the blood vessels in the present study were negative for human antigens. This may indicate a paracrine role of EPCs in angiogenesis by secreting different chemokines that promote recruitment of mouse resident cells, which will assumingly participate in blood vessel formation.

To investigate EPCs paracrine role in angiogenesis, we stained the resident mouse endothelial cells in the ectopic transplants, with mouse specific CD31 antibody. Mouse origin blood vessel count increased from ten days to three weeks in the EPC, and was higher from HNDF and β-TCP. These results suggest that endogenous EC were selectively recruited towards EPCs. Similar with our findings, Schuster et al., in a myocardial ischemia rat model, showed that the injection of EPC contributed to vessel formation by direct incorporation, but also due to paracrine effect of secreted proangiogenic factor [56]. Likewise, Cho et al. showed that human EPCs transplantation into the ischemic myocardium of mice increased the mobilization of BM-derived stem cells via the circulation system, and showed their incorporation into sites of vascularization and myocardial repair [57]. These results were also consistent with the study of Tongrong et al., which concluded that EPCs may exert a therapeutic effect in ischemic tissues by stimulating proliferation of surrounding cells via a paracrine effect [58]. Another study by Urbich showed that EPCs can enhance angiogenesis, by releasing cytokines, in particular VEGF and SDF-1, that promote the migration of ECs [59].

The common mesenchymal origin of endothelial cells and bone cells, and their close contact within the bone marrow, lead us to hypothesize that EPCs might transdifferentiate into osteoblasts. To test this theory, EPCs were cultured in osteogenic conditions, but failed to differentiate to bone forming cells (Appendix A). These results are consistent with those of Henrich [60], who investigated the influence of osteogenic culture conditions on EPCs metabolic activity and differentiation after seeding on the cancellous bone. Osteogenic stimulation of EPCs caused a decline, but not an abolishment, of endothelial characteristics, and did not induce osteogenic gene expression. In contrast, Han et al. showed that under osteogenic conditions, human EPCs lost their endothelial differentiation and expressed osteogenic genes [61]. In order to evaluate the ability of EPCs to recruit host MSCs, the ectopic tissues were stained with mouse CD73 antigens. In the EPC group, CD73 increased from ten days to three weeks; while in the other groups, CD73 levels were stable. At three weeks, CD73 levels were higher in the EPCs than the other groups. These results confirm the hypothesis that EPCs have a paracrine effect on MSC.

Since MSC are pluripotent and can differentiate into osteoblasts, chondrocytes, neurons, skeletal muscle cells, endothelial cells, and vascular smooth muscle cells [62,63], they are thought to contribute to neovascularization as well as osteogenesis. Recent studies suggest that EPCs contribute to MSCs osteogenic differentiation in vitro [64]. Aguirre et al. investigated the physical and biochemical interactions between bone marrow derived EPCs and MSCs in a co-culture system and demonstrated close cell-cell contacts soon after seeding. Tube formation was enhanced, even in starvation conditions, by the contribution of both MSCs and EPCs of bone marrow origin [65]. Moreover, Seebach et al. evaluated the influence of EPC, in combination with or without MSCs, on early revascularization and bone healing in a critical-sized defect (CSD), in vivo. The results showed significant improved vascularization and bone formation, and significantly bonier bridging in the EPCs/MSCs group. Based on these results, Seebach suggested a synergistic effect between EPCs and MSCs, and that the initial stage of revascularization/neovascularization by EPCs is considered crucial for complete bone regeneration in the late phase [44]. All these data suggest that a cross talk occurs between EPCs and the MSCs through paracrine and direct cell contact mechanisms.

To further validate the in vivo results, the paracrine effect of EPCs was investigated in vitro. EPC-CM stimulated the migration of MSCs and EPCs compared to standard medium. To understand the molecular mechanism behind this paracrine effect, quantitative real-time PCR analysis for 96 angiogenic genes was performed on EPC, that revealed high mRNA levels of angiogenic factors including: *FGF, KDR/VEGFR, CXCL12/SDF-1*, and osteogenic factors, such as: *PDGF* and *BMP-4.* In a previous study [23] we analyzed the mRNA expression of angiogenic and chemoattractant genes (*SDF-1*, *VEGF-A*, *CCL2*, *PDGFB*, *VEGFR-2* and *CXCR4* genes) in primary human EPCs isolated from ten unrelated donors. We found differences in RQ values between the donors, as can be expected for primary cells, and positive correlations between *SDF-1* and *CXCR4*.

Several studies showed that SDF-1 is a strong chemoattractant for CD34+ cells and MSC, which express CXCR4 (the receptor for SDF-1) [66,67] and stimulate the mobilization of EPCs [68] and MSCs [69,70,71], as well as circulating osteoblasts progenitor cells [72]. Indeed, in the current study, the addition of SDF-1 to standard medium stimulated MSCs and EPCs migration, while treating the cells with SDF-1 receptor antagonist (AMD3100) blocked cell migration. Nevertheless, the results of this study can’t rule out the possibility that SDF-1 binds to CXCR7. The migration of MSCs and EPCs towards 5ng/mL CCL2 (C-C-Motif Chemokine Ligand 2) was also tested. However, the results were less impressive compared to SDF-1 (Appendix A). In vivo studies supported these results; Otsuro et al. showed that vascular endothelial cells adjacent to ectopic bone expressed high levels of SDF-1. Bone marrow derived osteoprogenitors express CXCR4 and were mobilized to the ectopic site through the circulation system [72]. Salcedo et al. reported that subcutaneous serial injections of SDF-1 induced formation of local small blood vessels [73], while mice lacking SDF-1 had a defective formation of large vessels [66]. Liu et al. showed that treating mice suffering from femoral fracture with anti-SDF-1 caused the inhibition of new bone formation and fracture healing [74]. Overall, the cumulative data imply that SDF-1 secreted by endothelial cells plays a pivotal role in bone formation, by stimulating the mobilization of endothelial and osteoprogenitor cells from the bone marrow to the site of bone formation, via the SDF-1/CXCR4 pathway.

## 5. Conclusions

The results of our study indicate that the local transplantation of human EPCs seeded onto β-TCP scaffolds enhanced angiogenesis and osteogenesis in the ectopic transplants by direct engraftment to blood vessels, but more predominantly by the recruitment of endogenous cells. On the other hand, transplantation of HNDF showed no significant difference from a-cellular scaffold transplantation, suggesting that the paracrine activity that was observed is specific to EPCs. These results bolster our confidence that the role endothelial cells play in angiogenesis and osteogenesis is specific. Taken together, these findings indicate that osteogenesis and angiogenesis closely regulate each other in terms of micro-environmental interaction, with EPCs playing a significant paracrine role in these biological processes. Our future plans will include additional inquiries at the molecular level, to understand the pathways that EPCs interact with resident cells.

## Figures and Tables

**Figure 1 cells-09-01325-f001:**
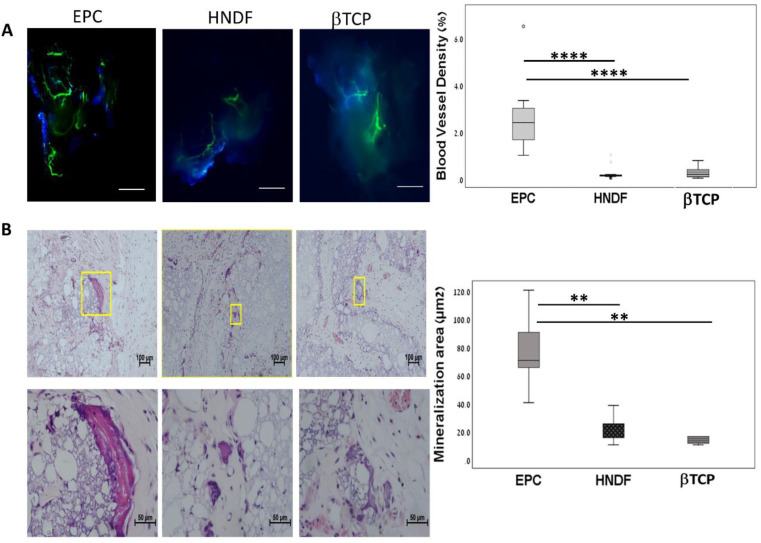
Endothelial progenitor cells (EPCs) stimulate angiogenesis and mineralization in ectopic subcutaneous mouse model. (**A**) Ten days after subcutaneous transplantation, blood vessels were stained with FITC dextran and visualized using LSM 510 Zeiss laser confocal system (Zeiss, Germany). Blood vessel density was quantified with IMARIS software (Portland, Oregon, USA). **** *p* < 0.0001. º means outlier, ^★^ means extreme outlier. Scale = 50 μm. (**B**) Eight weeks after transplantation, ectopic mineralization foci were observed in histological slides stained with H&E and quantified with Image Pro software (Rockville, MD, USA). ** *p* < 0.01. Scale = 100 μm (upper panel), 50 μm (lower panel).

**Figure 2 cells-09-01325-f002:**
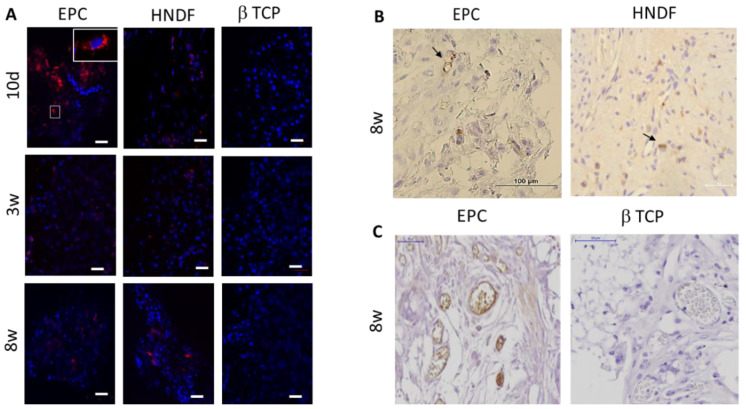
Engraftment of human EPCs in the ectopic transplants. (**A**) Human EPCs and human neonatal dermal fibroblasts (HNDF) were detected in the tissues (red fluorescent), up to eight weeks after transplantation. Scale = 50 μm. (**B**) Immunostaining with human nuclear antigen (HNA) was used to localize human HNDF and EPCs in the tissues (arrow). (**C**) Immunostaining with anti-human CD31 demonstrates integration of EPC into blood vessel walls.

**Figure 3 cells-09-01325-f003:**
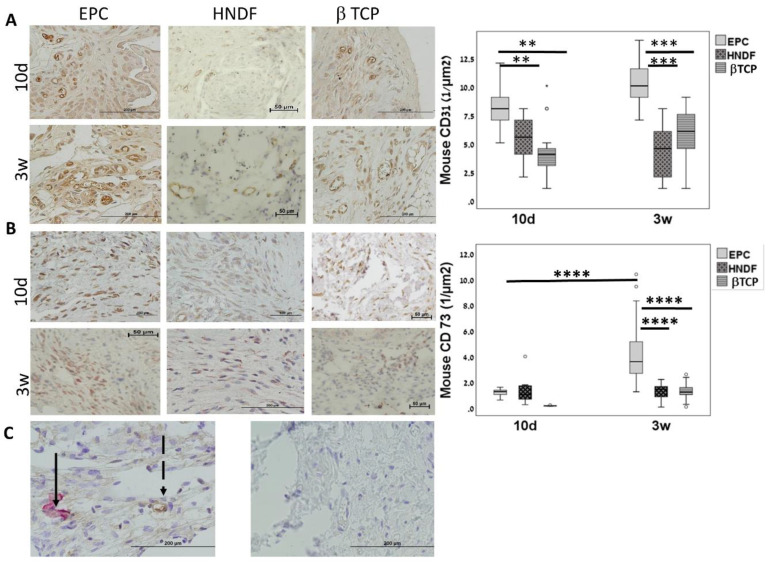
EPC recruit resident mesenchymal stem cells (MSCs) and ECs in vivo. (**A**) Recruitment of mouse CD31+ cells was assessed using anti mouse CD31 antibody. Immunostaining levels were quantified with Image Pro software (Rockville, MD, USA) ** *p* < 0.01. º means outlier, ^★^ means extreme outlier. Scale = 200 μm. (**B**) Recruitment of mouse CD73+ cells was assessed using anti mouse CD73 antibody. Immunostaining levels were quantified with Image Pro software (Rockville, MD, USA). **** *p* < 0.0001. º means outlier. Scale = 100 μm. (**C**) Double immunostaining with HNA antibody and anti-mouse CD73 of a representative slide from the EPC group showed proximity between the EPC (brown staining, dashed arrow) and mouse MSC (red staining, arrow). For negative control, double staining without primary antibodies was performed. Scale = 200 μm.

**Figure 4 cells-09-01325-f004:**
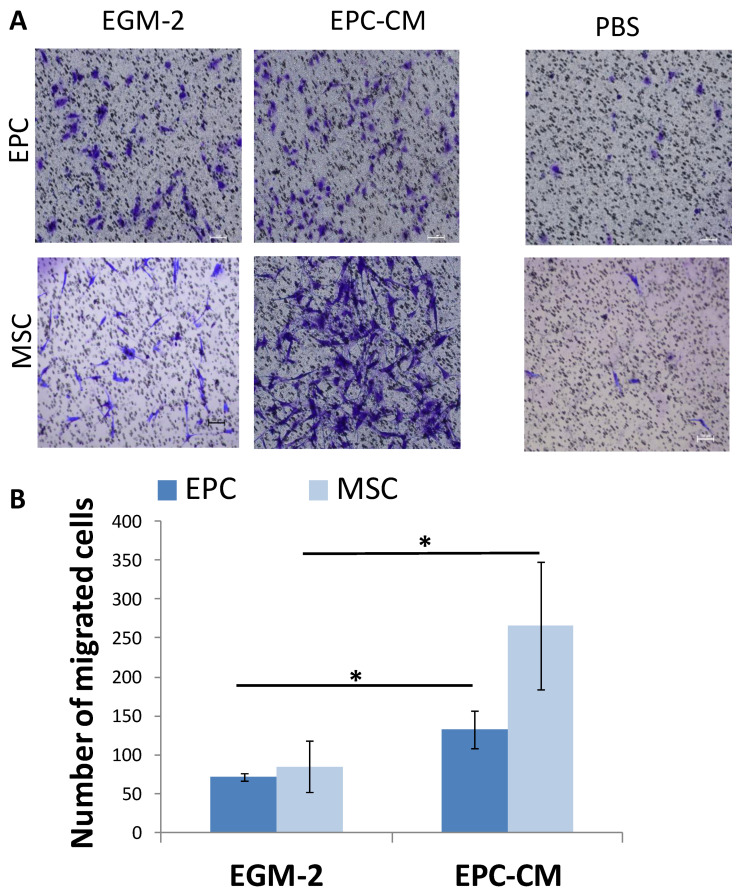
EPC-CM stimulate migration of MSC and EPC in Vitro. (**A**) Migration of EPCs and MSCs towards EPC-CM was evaluated in a transwell migration assay. For negative control, the lower chambers were filled with PBS. Scale = 100 μm. (**B**) Number of migrated MSCs and EPCs towards EPC-CM or EGM-2, * *p* < 0.05.

**Figure 5 cells-09-01325-f005:**
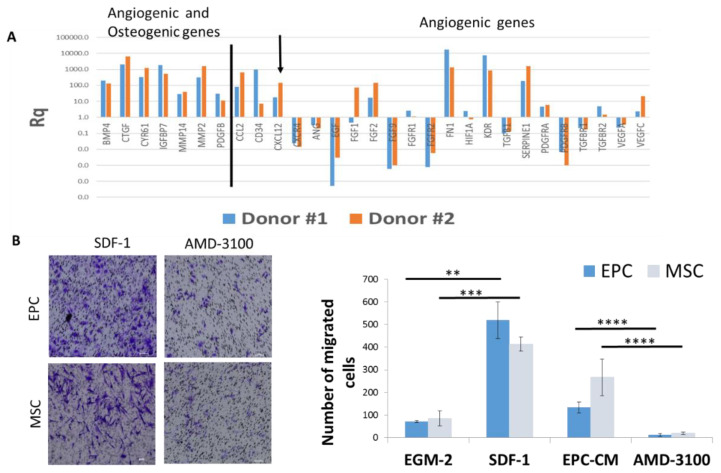
SDF-1 in EPC-CM plays a key role in EPC and MSC migration. (**A**) Relative quantification values of mRNA expression of osteogenic and angiogenic genes showed high expression of *CXCL12* (arrow). Results were normalized relative to housekeeping gene and MNCs from the same donors. RQ values obtained for all genes were added to Appendix A. (**B**) Migration of EPCs and MSCs towards SDF-1 was evaluated in a transwell migration assay. Migration was blocked by the addition of AMD3100 to the cells. ** *p* < 0.01, *** *p* < 0.001, **** *p* < 0.0001. Scale = 100 μm.

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
