# Peer review of "The Paracrine Role of Endothelial Cells in Bone Formation via CXCR4/SDF-1 Pathway"

_cells, 2020, doi:10.3390/cells9061325_

Round 1
Reviewer 1 Report
The article from Tamari and colleagues contains a clear description of the experiments performed to find out the pathways of endothelial progenitor cells in bone formation.
However I found a few minor modificatons that should be applied.
- the authors switch often between EPCs and hEPCs. Since they are talking about the same cell type (or at least that is what I understood), they should be more coherent and stick to one abbreviation.
- line 49: Mesenchymal Stem Cells (MSCs). The words stem and cells should start with a capital letter
- line 51: adipose derived MSCs. The underlined letters should be added
- line 53: Endothelial Progenitor Cells (EPCs). The underlined letters should be modified
- line 72: believe should be belief
- line 104: I wonder if there are markers to distinguish early and late EPCs. It would be easier to understand which pathway to search for
- lines 264-265 should be moved to Materials and Methods (M&M)
- from the pictures 3A and B it is very difficult to see the difference in the quantification and especially in picture 3A HNDF, it looks like the staining is increasing from 10d to 3w. Perhaps it would be more clear to use wider pictures with a close-up rectangle showing a particular of the staining
- lines 303, 305, 306 and 310: the abbreviations should be written in full first
- lines 353-358: this part should be moved to M&M
Reviewer 2 Report
RESULTS
- Please, indicate the number of ectopic mineralization foci analyzed in each group (Figure 1B) and remember to make graphs consistent using the (mu)greek letter and the correct superscript for exponents.
- Avoid if possible the use of two decimals in the "Y" axis scale.
- Figure 5A shows the expression of few genes from those present in the commercial plate has, at least as supplementary material the RQ values obtained for all genes would be welcome.
- Similarly, genes should be grouped by their main function, such as angiogenic or osteogenic.
- The analysis of this array of genes was only performed in two donors. It should be extended to a larger number, at least for the most significant genes found in the initial array, or at least for CXCL12/SDF-1.
- Why do the authors focus on SDF-1 and not on CCL2?, which is more expressed?
TYPOS
Altough centrifuge model was described, it is a good practice to describe centrifuge forces in (xg) rather that rpm, as these depend on rotor radius. In any case, please correct and be consistent.
Line 93: "750 rmp" for "750 rpm"
Line 175: "(250g, 5 min)" for "(250 xg, 5 min)"
Line 200: aKruskal Wallis change for "a Kruskal Wallis"
P statistic for P values should be italicized and capitalized.
According to the scientific guidelines for Human Gene nomenclature, Genes are italicized, and proteins are set in roman type (not italic). In humans both Capitalized.
Line 297: Scale=100um
Line: 321 Scale=100um
The greek letter for micro must be used i.e. 100μm
As common words "in vitro" and "in vivo" It is not necessary to be italicized, but hyphenation is incorrect.
Lines: 200-209-252-300-423-426.
Lines: 22-286-301.
Reviewer 3 Report
The study by Tamari and co-workers explores the role of endothelial cells in bone formation. This is an area that has been studied intensively as is acknowledged by the authors. The aim of this study was to provide insight into the mechanism how EPCs contribute to bone regeneration. EPCs have two modes of action, first they contribute to the cells within a vascular network and second, they secrete growth factors and cytokines to stimulate mesenchymal cells to differentiate into bone. In this study, the authors show that both mechanisms are involved in bone repair. In a paracrine manner the authors suggest that EPCs recruit host osteogenic cells using an ectopic bone model lacking osteoprogenitor cells.
- EPCs were isolated from post trauma hospitalized orthopedic patients and healthy donors. The authors mention that EPCs were cultured for 4 days after which non-adherent cells were removed. What is the time till first splitting the cells?
- It is not clear if the 3-5 pouches generated are filled with the same scaffold, or are the different conditions present in one mouse.
- Please provide the catalogue number of the antibodies used as different clones have different affinity/specificity
- Conditioned medium was prepared of cells cultured in EGM-2. It is not clear if serum was present as well. Furthermore, EGM-2 contains quite some pro-angiogenic cytokines. To make sure that the effect observed is not just due to the culture medium, EBM-2 should be used, or they should use EGM-2 as control.
- MSCs used: where are they derived from?
- As SDF-1 (I assume alpha?) can bind to different receptors, are the authors the effect seen is not due to binding to CXCR7?
- In figure 1 A the authors compare the ability of EPCs to stimulate angiogenesis in comparison to neonatal fibroblast. It is not of a surprise that EPCs are more angiogenic. More common used are MSCs. What is the rationale to use HNDF and not MSCs? Are MSCs more osteogenic?
- From the data presented it is not evident that EPCs are present in 2A after 10 days. Furthermore, while in figure 2B nuclear staining is clear for HNDF, the EPC staining is not convincing. Furthermore in figure 2C, the CD31 staining if bTCP is strange. Can the authors verify using an endothelial antibody recognizing both mouse and human ECs so we can appreciate the presence of cells, as is clear from 2A
- When analyzing figure 3A, it does see that HNDF has many more vessels compared to EPC, please explain the validation. Similar is the validation of figure 3B.
- In the migration assay, as EPCs are in their own medium they have an advantage compared to MSCs. Was this controlled for? If I compare the images in figure 4A this does not reflect the quantification.
- In figure 5, the authors show two donors, is there a big variation between donors, is there a difference in ability to isolate ECFCs?
- To really show that it is SDF-1, the authors should use and SDF1 neutralizing antibody. How does this relate to the study by Kawakami et al., 2014.
- It is not clear how EPCs from patients differ from healthy donors in their in vivo capacities.
Round 2
Reviewer 3 Report
The authors answered to all my questions with satisfaction. I have no more issues to address. I understand within the time frame given it is not possible to perform the sdf neutralizing experiment.